# VAEL: Bridging Variational Autoencoders and Probabilistic Logic Programming

**Eleonora Misino**
Department of Computer Science and Engineering
University of Bologna, Italy
`eleonora.misino2@unibo.it`

**Giuseppe Marra, Emanuele Sansone**
Department of Computer Science
KU Leuven, Belgium
`{first}.{last}@kuleuven.be`

## Abstract

We present VAEL, a neuro-symbolic generative model integrating variational autoencoders (VAE) with the reasoning capabilities of probabilistic logic (L) programming. Besides standard latent subsymbolic variables, our model exploits a probabilistic logic program to define a further structured representation, which is used for logical reasoning. The entire process is end-to-end differentiable. Once trained, VAEL can solve new unseen generation tasks by (i) leveraging the previously acquired knowledge encoded in the neural component and (ii) exploiting new logical programs on the structured latent space. Our experiments provide support on the benefits of this neuro-symbolic integration both in terms of task generalization and data efficiency. To the best of our knowledge, this work is the first to propose a general-purpose end-to-end framework integrating probabilistic logic programming into a deep generative model.

## 1 Introduction

Neuro-symbolic learning has gained tremendous attention in the last few years [4, 10, 33, 3] as such integration has the potential of leading to a new era of intelligent solutions, enabling the integration of deep learning and reasoning strategies (e.g. logic-based or expert systems). Indeed, these two worlds have different strengths that complement each other [31]. Deep learning systems excel at dealing with noisy and ambiguous high dimensional raw data, whereas reasoning systems leverage relations between symbols to reason and to generalize from a small amount of training data. While a lot of effort has been devoted to devising neuro-symbolic methods in the discriminative setting [47, 67, 49], less attention has been paid to the generative counterpart. An ideal generative neuro-symbolic framework should be able to encode the available small amount of training data into an expressive symbolic representation and to exploit complex forms of high level reasoning on such representation to generate new data samples. For example, consider a task where a single image of multiple handwritten numbers is labeled with their sum. Suppose that we want to generate new images not only given their addition, but also given their multiplication, power, etc. Common generative approaches, like VAE-based models, have a strong connection between the latent representation and the label of the training task (i.e. the addition) [35, 30]. Consequently, when considering new generation tasks that go beyond the simple addition, they have to be retrained on new data.

In this paper, we tackle the problem by providing a novel generative neuro-symbolic solution, named VAEL. In VAEL, the latent representation is not directly linked to the label of the task, but to a set

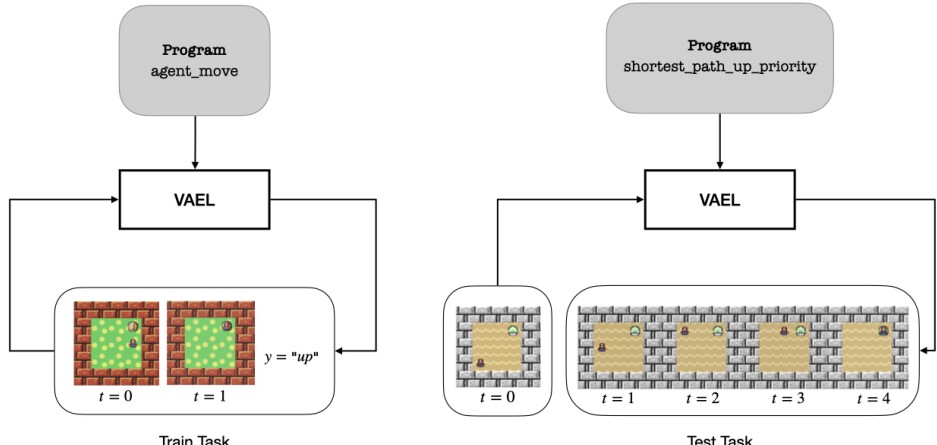

Figure 1: During training (**left**), VAEL leverages the symbolic structure of the input images provided by the ProbLog program *agent_move* to reconstruct both the images and the label. Once trained, VAEL can generalize to any new task involving reasoning with the same set of symbols (**right**) without retraining the model. This flexibility is achieved by replacing the ProbLog program used during training with the testing one. In the example, the program *shortest_path_up_priority* forces the agent in the input image ($t = 0$) to reach the target through the shortest path trajectory with priority on the *up* move.

of newly introduced symbols, i.e. logical expressions. Starting from these expressions, we use a *probabilistic logic program* to deduce the label. Importantly, the neural component only needs to learn a mapping from the raw data to this new symbolic representation. In this way, the model only weakly depends on the training data and can generalize to new generation tasks involving the same set of symbols (Figure 1). Moreover, the reasoning component offers a strong inductive bias, which enables a more data efficient learning.

The paper is structured as follows. In Section 2, we provide a brief introduction to probabilistic logic programming and to generative models conditioned on labels. In Section 3, we present the VAEL model together with its inference and learning strategies. Section 4 shows our experiments, while Section 5 places our model in the wider scenario of multiple related works. Finally, in Section 6, we draw some conclusions and discuss future directions.

## 2 Preliminaries

### 2.1 Probabilistic Logic Programming

A *logic program* is a set of *definite clauses*, i.e. expressions of the form $h \leftarrow b_1 \wedge ... \wedge b_n$, where $h$ is the *head literal* or conclusion, while the $b_i$ are *body literals* or conditions. Definite clauses can be seen as computational rules: IF all the body literals are true THEN the head literal is true. Definite clauses with no conditions ($n = 0$) are *facts*. In first-order logic programs, literals take the form $a(t_1, ..., t_m)$, with $a$ a predicate of arity $m$ and $t_i$ are the terms, that is constants, variables or functors (i.e. functions of other terms). Grounding is the process of substituting all the variables in an atom or a clause with constants.

ProbLog [9] lifts logic programs to *probabilistic logic programs* through the introduction of probabilistic facts. Whereas a fact in a logic program is deterministically true, a probabilistic fact is of the form $p_i :: f_i$ where $f_i$ is a logical fact and $p_i$ is a probability. In ProbLog, each ground instance of a probabilistic fact $f_i$ corresponds to an *independent Boolean random variable* that is true with probability $p_i$ and false with probability $1 - p_i$. Mutually exclusive facts can be defined through *annotated disjunctions* $p_0 :: f_0; \, ... \, ; p_n :: f_n$. with $\sum_i p_i = 1$. Let us denote with $\mathcal{F}$ the set of all ground instances of probabilistic facts and with $p$ their corresponding probabilities. Every subset $F \subseteq \mathcal{F}$ defines a *possible world* $w_F$ obtained by adding to $F$ all the atoms that can be derived from

$F$ using the logic program. The probability $P(w_F; p)$ of such a possible world $w_F$ is given by the product of the probabilities of the truth values of the probabilistic facts; i.e:

$$P(w_F; p) = \prod_{f_i \in F} p_i \prod_{f_i \in \mathcal{F} \setminus F} (1 - p_i) \tag{1}$$

Two inference tasks on these probabilities are of interest for this paper.

**Success**: The probability of a query atom $y$, or formula, also called *success probability of $y$*, is the sum of the probabilities of all worlds where $y$ is *True*, i.e.,

$$P(y; p) = \sum_{F \subseteq \mathcal{F}: w_F \models y} P(w_F; p) \tag{2}$$

**Sample with evidence**: Given a set of atoms or formulas $E$, the *evidence*, the probability of a world given evidence is:

$$P(w_F | E; p) = \frac{1}{Z} \begin{cases} P(w_F; p) & \text{if } w_F \models E \\ 0 & otherwise \end{cases} \tag{3}$$

where $Z$ is a normalization constant. Sampling from this distribution provides only worlds that are coherent with the given evidence.

**Example 1** (Addition of two digits). *Let us consider a setting where images contains two digits that can only be $0$ or $1$. Consider the following two logical predicates:* digit(img, I, Y) *states that a given image* img *has a certain digit* Y *in position* I, *while* add(img, z) *states that the digits in* img *sum to a certain value $z$.*

*We can encode the digit addition task in the following program $T$:*

```
p1 :: digit(img,1,0); p2 :: digit(img,1,1).
p3 :: digit(img,2,0); p4 :: digit(img,2,1).

add(img,Z) :- digit(img,1,Y1),
              digit(img,2,Y2),
              Z is Y1 + Y2.
```

*In this program $T$, the set of ground facts $\mathcal{F}$ is*

$$\{\texttt{digit}(\texttt{img}, 1, 0), \texttt{digit}(\texttt{img}, 1, 1), \texttt{digit}(\texttt{img}, 2, 0), \texttt{digit}(\texttt{img}, 2, 1)\}.$$

*The set of probabilities $p$ is $p = [p_1, p_2, p_3, p_4]$. The ProbLog program $T$ defines a probability distribution over the possible worlds and it is parameterized by $p$, i.e. $P(\omega_F; p)$. Then, we can ask ProbLog to compute the success probability of a query using Equation 2, e.g. $P(\texttt{add}(\texttt{img}, 1))$; or sample a possible world coherent with some evidence* add(img, 2) *using Equation 3, e.g. $\omega_F = \{\texttt{digit}(\texttt{img}, 1, 1), \texttt{digit}(\texttt{img}, 2, 1)\}$.*

## 2.2 Generation Conditioned on Labels

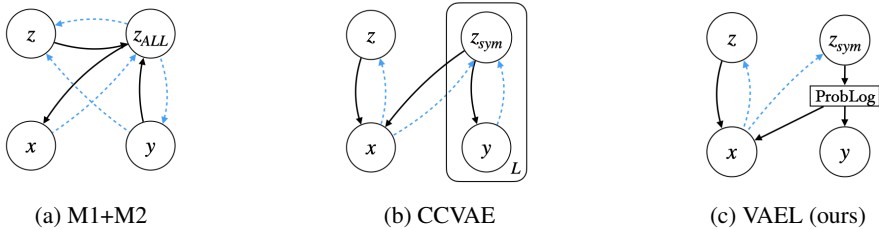

|(a) M1+M2 | (b) CCVAE | (c) VAEL (ours)|

Figure 2: Visual comparison for the probabilistic graphical models of [35] (M1+M2), of [30] (CCVAE) and ours (VAEL). Black arrows refer to the generative model, whereas blue dashed arrows correspond to the inference counterpart.

In this paper, we are interested in generative tasks where we consider both an image $x$ and a label $y$. The integration of supervision into a generative latent variable model has been largely investigated in

the past. For example, the work of [35] proposes an integrated framework between two generative models, called M1 and M2 (cf. Figure 2). Model M1 learns a latent representation for input $x$, i.e. $z_{ALL}$, which is further decomposed by model M2 into a symbolic and a subsymbolic vector $y$ and $z$, respectively. In this formulation, the generative process of the image is tightly dependent on the label, and therefore on the training task. More recently, another approach, called CCVAE [30], proposes to learn a representation consisting of two independent latent vectors, i.e. $z$ and $z_{sym}$, and forces the elements of $z_{sym}$ to have a one-to-one correspondence with the $L$ elements of $y$, thus capturing the rich information of the label vector $y$ (cf. Figure 2).

However, both the approaches are limited in terms of generation ability as their latent representation encodes information about the training task. This could be problematic when the label $y$ is only weakly linked to the true symbolic structure of the image. For example, let us consider the addition task in Example 1, where a single image of multiple handwritten numbers is labeled with their sum, e.g. $x = $ ⬛ and $y = 1$. In a generative task where we are interested in creating new images, using only the information of the label $y$ is not as expressive as directly using the values of the single digits. Moreover, suppose that we want to generate images where the two digits are related by other operations (e.g. subtraction, multiplication, etc). While we still want to generate an image representing a pair of digits, none of the models mentioned before would be able to do it without being retrained on a relabelled dataset. How can we overcome such limitations?

## 3  The VAEL Model

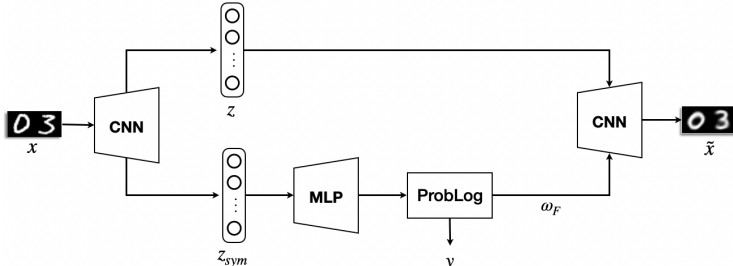

Figure 3: The VAEL model is composed of three components. First, the encoder (**left**) computes an approximated posterior of the latent variables $\mathbf{z}$ from the image $x$. The latent variables are split into two components: a subsymbolic $z$ and a symbolic $z_{sym}$. Second, $z_{sym}$ is used to parameterize a ProbLog program (**center**). A MLP is used to map the real variables $z_{sym}$ into the probabilities of the facts in the program. Then, the program is used to compute the label $y$ and a possible world $\omega_F$. Finally, a decoder (**right**) takes both the latent vector $z$ and the possible world from ProbLog to reconstruct the image $\tilde{x}$.

Here, we propose a probabilistic graphical model which enables to unify VAEs with Probabilistic Logic Programming. The graphical model of VAEL (Figure 2) consists of four core variables. $x \in \mathbb{R}^{H \times W \times C}$ represents the image we want to generate, while $y \in \{0, 1\}^K$ represents a label, i.e. a symbolic information characterizing the image. The latent variable is split into a symbolic component $z_{sym} \in \mathbb{R}^N$ and a subsymbolic component $z \in \mathbb{R}^M$. Conversely to other VAE frameworks, VAEL does not rely on a one-to-one mapping between $y$ and $z_{sym}$, rather it exploits a probabilistic logic program to link them. Indeed, the probabilistic facts $\mathcal{F}$ are used by the ProbLog program $T$ to compute the actual labels $y$ and they can encode a more meaningful symbolic representation of the image than $y$.

**Generative model.**

The generative distribution of VAEL (Figure 2) is factorized in the following way:

$$p_\theta(x, y, \mathbf{z}) = p(x|\mathbf{z})p(y|z_{sym})p(\mathbf{z}) \tag{4}$$

where $\mathbf{z} = [z_{sym}, z]$ and $\theta$ are the parameters of the generative model. $p(\mathbf{z})$ is a standard Gaussian distribution, while $p(y|z_{sym})$ is the success distribution of the label of the ProbLog program $T$ (Eq. 2). $p(x|\mathbf{z})$ is a Laplace distribution with mean value $\mu$ and identity covariance, i.e. $Laplace(x; \mu, I)$. Here, $\mu$ is a neural network decoder whose inputs are $z$ and $\omega_F$. $\omega_F$ is sampled from $P(\omega_F; MLP(z_{sym}))$ (Eq. 1).

**Inference model.** We amortise inference by using an approximate posterior distribution $q_\phi(\mathbf{z}|x, y)$ with parameters $\phi$. Furthermore, we assume that $\mathbf{z}$ and $y$ are conditionally independent given $x$, thus obtaining $q_\phi(\mathbf{z}|x, y) = q_\phi(\mathbf{z}|x)$[1]. This allows us to decouple the latent representation from the training task. Conversely, the other VAE frameworks do not exploit this assumption and have a latent representation that is dependent on the training task.

The overall VAEL model (including the inference and the generative components) is shown in Figure 3.

**Objective Function.** The objective function of VAEL computes an evidence lower bound (ELBO) on the log likelihood of pair $(x, y)$, namely:

$$\mathcal{L}(\theta, \phi) = \mathcal{L}_{REC}(\theta, \phi) + \mathcal{L}_Q(\theta, \phi) - \mathcal{D}_{\mathcal{KL}}[q_\phi(\mathbf{z}|x)||p(\mathbf{z})]] \tag{5}$$

where

$$\mathcal{L}_{REC}(\theta, \phi) = \mathbb{E}_{\mathbf{z} \sim q_\phi(\mathbf{z}|x)}[\log(p(x|\mathbf{z})], \quad \mathcal{L}_Q(\theta, \phi) = \mathbb{E}_{z_{sym} \sim q_\phi(z_{sym}|x)}[\log(p(y|z_{sym}))]].$$

Note that we omit the dependence on $\omega_F$ in the objective, thanks to an equivalence described in the extended derivation (see Appendix **??**).

The objective is used to train VAEL in an end-to-end differentiable manner, thanks to the Reparametrization Trick [34] at the level of the encoder $q_\phi(\mathbf{z}|x)$ and the differentiability of the ProbLog inference, which is used to compute the success probability of a query and sample a world.

In Appendix **??** we report VAEL training algorithm (Algorithm **??**) along with further details on the training procedure.

### 3.1 Downstream Applications

**Label Classification.** Given $x$ we use the encoder to compute $z_{sym}$ and by using the MLP we compute the probabilities $p = MLP(z_{sym})$. Then, we can predict labels by computing the probability distribution over the labels $P(y; p)$, as defined in Eq. 2, and sampling $y \sim P(y; p)$. This process subsumes the DeepProbLog framework [47].

**Image Generation.** We generate images by sampling $\mathbf{z} = [z_{sym}, z]$ from the prior distribution $\mathcal{N}(0, 1)$ and a possible world $\omega_F$ from $P(\omega_F; p)$. The distribution over the possible worlds $P(\omega_F; p)$ is computed by relying on ProbLog inference starting from the facts probabilities $p = MLP(z_{sym})$.

**Conditional Image Generation.** As described in Section 2.1, ProbLog inference allows us also to *sample with evidence*. Thus, once sampled $\mathbf{z}$ from the prior, we can (i) compute $p = MLP(\hat{z}_{sym})$, then (ii) compute the conditional probability $P(\omega_F \mid E; p)$, (iii) sampling $\omega_F \sim P(\omega_F \mid E; p)$ and (iv) generate an image consistent with the evidence $E$.

**Task Generalization.** As we have seen, VAEL factorizes the generation task into two steps: (i) generation of the world $\omega_F$ (e.g. the digits labels); (ii) generation of the image given the world. Whereas the second step requires to be parameterized by a black-box model (e.g. a convolutional neural network), the generation of a possible world $\omega_F$ is handled by a symbolic generative process encoded in the ProbLog program $T$. Thus, once trained VAEL on a specific symbolic task (e.g. the addition of two digits), we can generalize to any novel task that involves reasoning with the same set of probabilistic facts by simply changing the ProbLog program accordingly (e.g. we can generalize to the multiplication of two integers). To the best of our knowledge, such a level of task generalization cannot be achieved by any other VAE frameworks.

## 4 Experiments

In this Section, we validate our approach on the four downstream applications by creating two different datasets. Finally, we provide an analysis in terms of data efficiency.

**2digit MNIST dataset.** We create a dataset of $64, 400$ images of two digits taken from the MNIST dataset [38]. We use 65%, 20%, 15% splits for the train, validation and test sets, respectively. Each image in the dataset has dimension $28 \times 56$ and is labelled with the sum of the two digits. The dataset contains a number of images similar to the standard MNIST dataset. However, it is combinatorial in nature, making any task defined on it harder than its single-digit counterpart.

---

[1] We use a Gaussian distribution with a mean parameterized by the encoder network and identity covariance

**2level Mario dataset.** We create a dataset containing $6,720$ images of two consequent states of a $3 \times 3$ grid world where an agent can move by one single step (diagonals excluded). Each image has dimension $100 \times 200$ and is labelled with the move performed by the agent. For example, the image in Figure 4 has label down. We use $70\%$, $20\%$, $10\%$ splits for the train, validation and test sets, respectively.

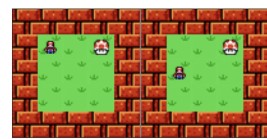

Figure 4: Example of *2level Mario* dataset image. The $3 \times 3$ grid world (green area) is surrounded by a frame (bricks).

Both datasets present a compositional scene that showcases the advantages of using a neuro-symbolic approach to logically reasoning upon the scene components. Moreover, the compositional nature allows us to design novel tasks that imply arbitrarily complex forms of reasoning among the scene elements, thus testing the generalization capability of the models.

**Evaluation.** In order to evaluate our approach, we rely on a *reconstruction loss* ($m_{REC}$) in terms of data log-likelihood and two accuracies, *predictive* ($m_{CLASS}$) and *generative* ($m_{GEN}$). Regarding the *predictive accuracy*, we measure the predictive ability of the model as the classification accuracy on the true labels (the *addition* of the two digits for *2digit MNIST* dataset, and the *move* for *2level Mario* dataset). It is worth mentioning that, for *2digit MNIST* dataset, such accuracy cannot be directly compared with standard values for the single-digit MNIST, as the input space is different: the correct classification of an image requires both the digits to be correctly classified. The *generative accuracy* is assessed by using an independent classifier for each dataset. For *2digit MNIST* dataset, the classifier is trained to classify single digit value; while for the *2level Mario* dataset, the classifier learns to identify the agent's position in a single state. The evaluation process for the generative ability can be summarized as: (i) jointly generate the image and the label $\tilde{y}$; (ii) split the image into two sub-images and (iii) classify them independently; (iv) finally, for *2digit MNIST* dataset, we sum together the outputs of the classifier and we compare the resulting addition with the generated label $\tilde{y}$; while for *2level Mario* dataset, we verify whether the classified agent's positions are consistent with the generated label $\tilde{y}$.

In the following tasks, we compare VAEL against CCVAE [30] when possible. In order to make a fair comparison, we clearly state the differences between the two models. VAEL can be applied to any conditional generative task. In those tasks where no extra information on the compositional nature of the scene is available, we can design the ProbLog program to encode only the label $y$, and VAEL collapses to CCVAE. Whenever we can access extra information to map $y$ to a more fine-grained symbolic representation, we can inject this knowledge into the model through a ProbLog program. VAEL exploits the additional information to improve task generalization and data efficiency, as we will show in this section. Conversely, CCVAE is not able to exploit all the available information, thus requiring more training data to perform comparably and preventing reaching the same level of task generalization. The source code and the datasets are available at `https://github.com/elemisi/vael` under MIT license. Further implementation details can be found in Appendix **??**.

**Label Classification.** In this task, we want to predict the correct label given the input image, as measured by the predictive accuracy $m_{CLASS}$. Both VAEL and CCVAE use an encoder to map the input image to a latent vector $z_{sym}$. VAEL uses ProbLog inference to predict the label $y$. In contrast, CCVAE relies on the distribution $p(y|z_{sym})$, which is parameterized by a neural network. As shown in Table 1, CCVAE and VAEL achieve comparable predictive accuracy in *2level Mario* dataset. However, VAEL generalizes better than CCVAE in *2digit MNIST* dataset. The reason behind this performance gap is due to the fact that the *addition* task is combinatorial in nature and CCVAE would require a larger number of training samples in order to solve it. We further investigate this aspect in the *Data efficiency* experiment.

**Image Generation.** We want to test the performance when generating both the image and the label. VAEL generates both the image and the label $\tilde{y}$ starting from the sampled latent vector $\mathbf{z} \sim \mathcal{N}(0,1)$. Conversely, CCVAE starts by sampling the label $\tilde{y}$ from its prior, then proceeds by sampling the latent vector from $p(\mathbf{z}|y = \tilde{y})$, and finally generates the new image. Figure 5a shows some random samples for both models for *2digit MNIST* dataset. The pairs drawn by VAEL are well defined, while CCVAE generates more ambiguous digits (e.g., the $1$ resembles a $0$, the $4$ may be interpreted as a $9$, and so on). This ambiguity makes it harder for the classifier network to distinguish among the digits during the evaluation process, as confirmed by the quantitative results in Table 1, where VAEL outperforms

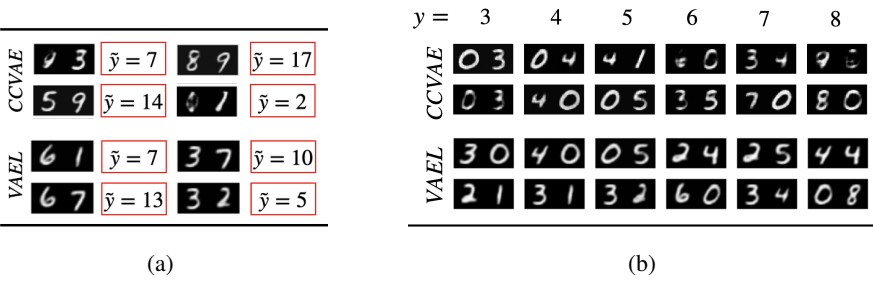

(a)          (b)

Figure 5: Examples of generation (a) and conditional generation (b) for VAEL and CCVAE on *2digit MNIST* dataset. In (b) in each column the generation is conditioned on a different label $y$.

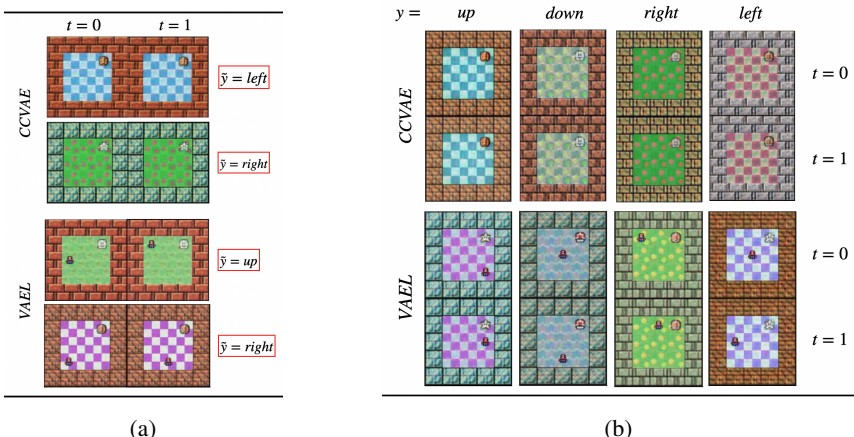

(a)          (b)

Figure 6: Examples of generation (a) and conditional generation (b) for VAEL and CCVAE on *2level Mario* dataset. In (b) in each column the generation is conditioned on a different label $y$.

CCVAE in terms of generative ability. Regarding *2level Mario* dataset (Figure 6a), VAEL is able to generate data-like images, where the background is preserved from one state to the subsequent one (additional results can be found in Appendix **??**). Conversely, CCVAE fails the generation task: although it correctly generates the background, it is not able to draw the agent. This is also supported by the disparity in the reconstructive ability, as reported in Table 1. In *2level Mario* dataset, this is due to a systematic error in which CCVAE focuses only on reconstructing the background, thus discarding the small portion of the image containing the agent, as shown in Figures 6a, 6b and in Appendix **??**. The difference in performance between CCVAE and VAEL lies in the fact that for each label there are many possible correct images. For example, in the 2level Mario dataset, there are 6 possible pairs of agent's positions that correspond to the label `left`. Our probabilistic logic program explicitly encodes the digits value or the single agent's positions in its probabilistic facts, and uses the variable $z_{sym}$ to compute their probabilities. On the contrary, CCVAE is not able to learn the proper mapping from the digits value or the agent's positions to the label, but it can learn to encode only the label in the latent space $z_{sym}$.

**Conditional Image Generation.** In this task, we want to evaluate also the conditional generation ability of our approach. In Figures 5b and 6b we report some qualitative results for both VAEL and CCVAE (additional results can be found in Appendix **??**). As it can be seen in 5b, VAEL always generates pairs of digits coherent with the evidence, showing also a variety of combinations. Conversely, some of the pairs generated by CCVAE do not sum to the desired value. Regarding *2level Mario* dataset (Figure 6b), VAEL generates pairs of states coherent with the evidence, and with different backgrounds that are preserved from one state to the subsequent one. On the contrary, CCVAE is not able to draw the agent in the generated images, thus failing the task. The reason lies, again, in the task complexity, that VAEL reduces by relying on its probabilistic logic program.

Table 1: Reconstructive, predictive and generative ability of VAEL and CCVAE. We use repeated trials to evaluate both the models on a test set of $10K$ images for *2digit MNIST* dataset and 1344 images for *2level Mario* dataset.

| Dataset | Model | $m_{REC}(\downarrow)$ | $m_{CLASS}(\uparrow)$ | $m_{GEN}(\uparrow)$ |
|---|---|---|---|---|
| *2digit MNIST* | CCVAE | $1549 \pm 2$ | $0.5284 \pm 0.0051$ | $0.5143 \pm 0.0157$ |
| | VAEL | $\mathbf{1542 \pm 3}$ | $\mathbf{0.8477 \pm 0.0178}$ | $\mathbf{0.7922 \pm 0.0350}$ |
| *2level Mario* | CCVAE | $43461 \pm 209$ | $\mathbf{1.0 \pm 0.0}$ | $0.0 \pm 0.0$ |
| | VAEL | $\mathbf{42734 \pm 246}$ | $\mathbf{0.977 \pm 0.0585}$ | $\mathbf{0.8135 \pm 0.2979}$ |

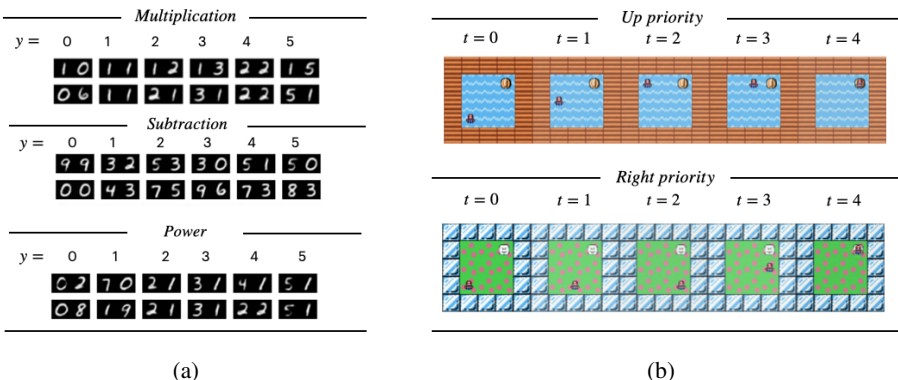

(a)                                              (b)

Figure 7: Examples of the generation ability of VAEL in previously unseen tasks for *2digit MNIST* dataset (a) and *2level Mario* dataset (b).

**Task Generalization.** We define several novel tasks to evaluate the task generative ability of VAEL. For *2digit MNIST* dataset, we introduce the *multiplication*, *subtraction* and *power* between two digits, while for *2level Mario* dataset we define two shortest paths (*up priority*, i.e. up always first, and one with *right priority*, i.e. right always first). To the best of our knowledge, such a level of task generalization cannot be achieved by any existing VAE framework. On the contrary, in VAEL, we can generalize by simply substituting the ProbLog program used for the training task with the program for the desired target task, without re-training the model. In Figure 7, we report qualitative results: in 7a, the generation is conditioned on a different label $y$ referring to the corresponding mathematical operation between the first and second digit; in 7b, the model is asked to generate a trajectory starting from the initial image ($t = 0$) and following the shortest path using an *up priority* or a *right priority*.

In all the novel tasks of *2digit MNIST* dataset (Figure 7a), VAEL generates pairs of numbers consistent with the evidence, and it also shows a variety of digits combinations by relying on the probabilistic engine of ProbLog. This should not surprise. In fact, in all these tasks, the decoder takes as input a possible world, i.e., a specific configuration of the two digits. Therefore, the decoder is agnostic to the specific operation, which is entirely handled by the symbolic program. For this reason, VAEL can be seamlessly applied to all those tasks that require the manipulation of two digits. The same reasoning can be extended to *2level Mario* novel tasks (Figure 7b), where VAEL generates subsequent states consistent with the shortest path, while preserving the background of the initial state ($t = 0$) thanks to the clear separation between the subsymbolic and symbolic latent components. Additional results can be found in Appendix **??**.

**Data Efficiency.** In this task, we want to verify whether the use of a logic-based prior helps the learning in contexts characterized by data scarcity. To this goal, we define different training splits of increasing size for the *addition* task of *2digit MNIST* dataset. In particular, the different splits range from 10 up to 100 images per pair of digits. The results (Figure 8) show that VAEL outperforms the baseline for all the tested sizes. In fact, with only 10 images per pair, VAEL already performs better than CCVAE trained with 100 images per pair. When considering 10 images per pair, the discriminative and generative accuracies of VAEL are $0.445 \pm 0.057$ and $0.415 \pm 0.0418$, whereas CCVAE trained on 100 images per pair has a discriminative and generative accuracy of $0.121 \pm 0.006$

and $0.284 \pm 0.006$ respectively. The reason behind this disparity is that the logic-based prior helps the neural model in properly structuring the latent representation, so that one part can easily focus on recognizing individual digits and the other on capturing the remaining information in the scene. Conversely, CCVAE needs to learn how to correctly model very different pairs that sum up to the same value. We further investigated the gap between CCVAE and VAEL by running the same experiment in a simplified setting with only three possible digits values: 0, 1 and 2. CCVAE bridges the gap with VAEL only when trained with a dataset of 24,000 images, which solidifies VAEL's advantage in data-scarce settings. Additional details can be found in Appendix **??**.

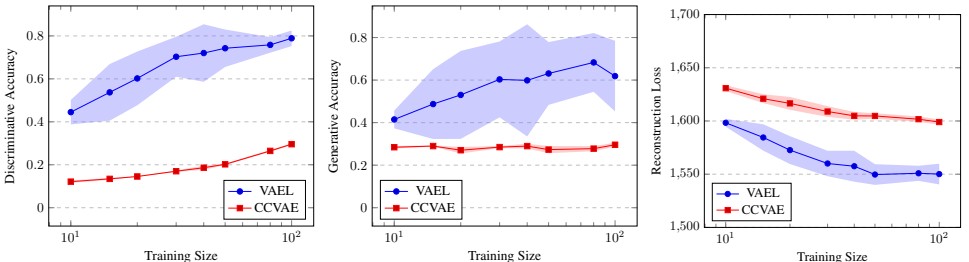

Figure 8: Discriminative, generative and reconstructive ability of VAEL (**blue**) and CCVAE (**red**) trained in contexts characterized by data scarcity. Both the models are evaluated on the same test set. The training size refers to the number of samples per pair of digits seen during the training.

## 5  Related Work

**Controlled image generation**. We distinguish between generative models based on text descriptions and generative models based on scene graphs. Regarding the first category, substantial effort has been devoted to devising strategies able to generate images with control (i) on object properties/attributes (e.g. shape, color, texture of objects) [55, 56, 68, 69, 13], (ii) on spatial relations between multiple objects (e.g. object A is below object B) [48, 52, 24, 44], (iii) or both [53]. Our framework is related to these works as considering the problem of generation in a relational setting. Differently from them, we use probabilistic logic programming to encode first-order logical knowledge and to perform reasoning over this knowledge. This comes with the advantage that we can generalize to out-of-distribution relations, which consists of both the composition of previously seen relations (e.g. the multiplication can be composed by using the sum in the domain of natural numbers) and new relations (e.g. the subtraction cannot be composed by using the sum in the domain of natural numbers). Regarding the second category, scene graphs are used as an alternative to text descriptions to explicitly encode relations, such as spatial relations between objects [29, 1, 22, 40, 50, 23, 25, 6]. While related, our approach differs from these last as logical programs are more expressive and allow a more general reasoning than scene graphs alone.

**Unsupervised scene decomposition** We distinguish between object-oriented, part-oriented and hierarchical approaches. The first category attempts to learn individual object representations in an unsupervised manner and to reconstruct the original image or the subsequent frame (in the case of sequential data) from these representations. Several approaches have been proposed, based on scene-mixtures [19, 61, 5, 20, 14, 45, 36, 60], spatial attention models [21, 15, 8] and their corresponding combination [43, 28]. In the second category, a scene with an object is decomposed into its constituent parts. Specifically, an encoder and a decoder are used to decompose an object into its primitives and to recombine them to reconstruct the original object, respectively. Several approaches have been proposed for generating 3D shapes [63, 39, 70, 26, 32, 11] and for inferring the compositional structure of the objects together with their physical interactions in videos [66, 41, 17]. These approaches focus on learning the part-whole relationships of object either by using pre-segmented parts or by using motion cues. Last but not least, there has been recent effort focusing on integrating the previous two categories, thus learning to decompose a scene into both its objects and their respective parts, the so called hierarchical decomposition [57, 12]. Our work differs in several aspects and can be considered as an orthogonal direction. First of all, we consider static images and therefore we do not exploit temporal information. Secondly, we do not provide any information about the location of the objects or their parts and use a plain autoencoder architecture to discover the objects. Therefore, we could exploit architectural advances in unsupervised scene decomposition to

further enhance our framework. However, this integration is left to future investigation. Finally, our model discovers objects in a scene, by leveraging the high-level logical relations among them.

**Neuro-symbolic generation**. This is an emerging area of machine learning as demonstrated by works appeared in the last few years. For example, [27] proposes a generative model based on a two-layered latent representation. In particular, the model introduces a global sub-symbolic latent variable, capturing all the information about a scene and a symbolic latent representation, encoding the presence of an object, its position, depth and appearance. However, the model is limited in the form of reasoning, as able to generate images with objects fulfilling only specific spatial relations. In contrast, our model can leverage a logical reasoning framework and solve tasks requiring to manipulate knowledge to answer new generative queries.

There are two recent attempts focusing on integrating generative models with probabilistic programming [16, 18], where reasoning is limited to spatial relationships of (parts of) the image. Moreover, [18] is a clear example of the difficulty of integration the symbolic and the perceptual module. In contrast, our work provides a unified model which can learn to generate images while perform logical reasoning at the same time.

To the best of our knowledge, the work in [58] represents the first attempt to integrate a generative approach with a logical framework. However, the work differs from ours in several aspects. Firstly, the authors propose a model for an image completion problem on MNIST and it is unclear how the model can be used in our learning setting and for generating images in the presence of unseen queries. Secondly, the authors propose to use sum-product networks as an interface between the logical and the neural network modules. In contrast, we provide a probabilistic graphical model which compactly integrates the two modules without requiring any additional network. Thirdly, we are the first to provide experiments supporting the benefits of such integration both in terms of task generalization and data efficiency.

**Structured priors for latent variable models**. Several structured priors have been proposed in the context of latent variable models. For example, The work in [62] focuses on learning priors based on mixture distributions. [2] uses rejection sampling with a learnable acceptance function to construct a complex prior. The works of [59, 46, 64, 37] consider learning hierarchical priors, [7, 51, 54] introduce autoregressive priors [7]. While structured priors offer the possibility of learning flexible generative models and avoid the local minima phenomenon observed in traditional VAEs, they are quite different from ours. Indeed, our prior disentangles the latent variables to support logical reasoning. Furthermore, the structure of the logic program is interpretable.

## 6 Conclusions and Future Works

In this paper, we presented VAEL, a neuro-symbolic generative model that integrates VAE with Probabilistic Logic Programming. The symbolic component allows to decouple the internal latent representation from the task at hand, thus allowing an unprecedented generalization power. We showcased the potential of VAEL in two image generation benchmarks, where VAEL shows state-of-the-art generation performance, also in regimes of data scarcity and in generalization to several prediction tasks.

In the future, we plan to improve VAEL by i) investigating alternative and more scalable semantics for probabilistic programs (e.g. stochastic logic program [65]), and ii) exploring different solutions to achieve out-of-distribution generalization. Moeover, we plan to apply VAEL to other settings, like structured object generation [42], to showcase the flexibility and expressivity provided by the integration with a probabilistic logic program.

## Acknowledgements

GM is funded by the Research Foundation-Flanders (FWO-Vlaanderen, GA No 1239422N). This research is funded by TAILOR, a project funded by EU Horizon 2020 research and innovation programme under GA No 952215. ES is partially funded by the KU Leuven Research Fund (C14/18/062) and the Flemish Government (AI Research Program). The authors would like to thank Luc De Raedt for supporting this project as an Erasmus Master Thesis, and Federico Ruggeri for his support in the experimental phase.

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
