# OpenReview forum: "VAEL: Bridging Variational Autoencoders and Probabilistic Logic Programming"
_NeurIPS.cc/2022/Conference — NeurIPS 2022 Accept_

### Official Review · Reviewer_2dcW · 2022-07-05

**Rating:** 6
**Confidence:** 3
**Soundness:** 3 good
**Presentation:** 4 excellent
**Contribution:** 3 good

**Summary:**

This paper proposes to integrate VAE with ProbLog to take advantage of the extra symbolic-program information for conditional image generation with better data efficiency and generalizability.

**Questions:**

* How does the model perform on more realistic datasets such as ARC or CLEVR?

**Limitations:**

The paper mentions some limitations such as the scalability issue of ProbLog and the simplicity of the tasks. I would strongly recommend the authors mention the difference in settings when compared with traditional conditional image generation methods such as CCVAE.

**Strengths And Weaknesses:**

This paper studies an interesting field of neuro-symbolic methods, conditional generation. The method, which incorporates ProbLog to map weak image labels to decomposed symbolic representations in a differentiable way given the correct program, is straightforward and intuitive. Experimental results support their claims on two synthetic datasets, showing that their method can achieve better data efficiency and task generalizability with the help of the information of symbolic programs. The paper is well written and easy to understand.

There is a major concern:
* The datasets are simple and synthetic. There exist more complicated choices such as ARC [1], CLEVR, or some other more realistic VQA datasets. Their images are also associated with structures or world descriptions, while the tasks are more complex and more difficult.

as well as some other concerns:
* It's unclear if the setting used in this paper is practical. They need extra information about the symbolic representations (how to model facts in ProbLog, e.g., 3x3 grids) and the label deduction program (how to infer labels from facts). This is different from the classical conditional generation setting.
* Meanwhile, the authors should clearly state the difference in settings while comparing with CCVAE, especially about the extra information incorporated by VAEL, for a fair comparison and evaluation. Specifically, for the Mario dataset, it is very difficult to learn the program in the general conditional image generation setting because there are only 4 classes/labels, each with 6 pairs of states, resulting in only 24 symbolic data points in the whole dataset.  VAEL, on the other hand, is directly given the correct program, and the setting is thus much easier.

Nevertheless, it is still interesting to see methods integrating logical reasoning into conditional generative models and demonstrating better data efficiency and task generalizability. In the long term, it would be more realistic and exciting to consider jointly learning the symbolic representations and the programs with other modules. I would be more convinced and increase my score when provided with experimental results on more complex and realistic datasets.

[1] Chollet, François. "On the measure of intelligence." arXiv preprint arXiv:1911.01547 (2019).

---

> ### Author Response · Authors · 2022-08-02
> **Answer to Reviewer 2dcW**
>
> We thank the reviewer for their positive assessment, encouraging words, and for suggesting interesting future work directions. In the following sections, we discuss the main concerns expressed by the reviewer.
>
> ## Datasets
> In this work, we focus on the least non-trivial datasets to simplify the analysis of our new system. The compositional nature of the datasets we propose allows us to test the fundamental properties of our approach. Moreover, the two datasets showcase the advantages of relying on a neuro-symbolic integration: despite their simplicity, the two datasets are yet challenging for the state-of-the-art CCVAE. As a final remark, we thank the reviewer for suggesting us two interesting and challenging datasets, which represent a promising future research direction to further apply VAEL in compositional tasks.
>
> ## Use of extra information
>
> Regarding the practicability of our method, the extra information is nothing more than a formal specification of the desired properties of the scene, which provides a formal structure in the latent space. The expressivity of the ProbLog program allows the user to compactly represent the formal specifications in terms of first-order logic rules. Therefore, conversely to propositional logic, there is no need to specify the grounding over the elements of the Herbrand base. Hence, our setting minimizes the human effort required to inject the logical constraints into the model.
>
> ## Comparison with CCVAE
> It is worth noting that both the tasks we designed can be analyzed in terms of classic conditional image generation tasks. For example, in MARIO, given the label of a transition, we generate two images of consecutive game states with that transition. The novelty of these tasks is that the label can be mapped to a more fine-grained symbolic representation. However, such mapping is not deterministic; e.g., we have multiple pairs of states for the same transition. This aspect motivates the need for a language for such symbolic stochastic mappings: a probabilistic logic program. Concerning CCVAE, note that nothing prevents it from learning the mapping and succeeding in this task, but it has to do it in a subsymbolic way. Thus, the fundamental issue is that, to the best of our knowledge, there is no way for CCVAE, or any other conditional generative model, to exploit a known stochastic symbolic mapping.

---

> > ### Comment · Reviewer_2dcW · 2022-08-05
> > **Quick reply to authors**
> >
> > I would like to thank the authors for their reply to my review.
> >
> > As a quick reply to the authors, I would emphasize the need for a clear statement of the difference between their method V.S. CCVAE, etc. Otherwise, the paper may not be accurate enough for publication.
> > * As stated in the authors' own rebuttal, their methods use extra information such as (1) "the label can be mapped to a more fine-grained symbolic representation", (2) "the human effort required to inject the logical constraints into the model", and (3) what the symbolic representations are, according to my understanding.
> > * The extra information may not always be available for general practical conditional image generation tasks, which shows the limitation of the applicability of the proposed method. On the other hand, classical methods don't have these limitations.
> >
> > Even though I still appreciate the proposed method for its ability to take advantage of the extra symbolic information, I would strongly suggest authors adjust their presentations for accurate descriptions and fair comparison.

---

> > > ### Author Response · Authors · 2022-08-06
> > > **Reply to Reviewer 2dcW**
> > >
> > > We thank the reviewer for their feedback. We will include an additional statement to highlight the difference between CCVAE and VAEL in terms of extra information.
> > > In particular, we will add the following paragraph on line 189:
> > >
> > > *"In order to make a fair comparison, we clearly state the differences between VAEL and the baseline. VAEL can be applied to any conditional generative task. In those tasks where no extra information on the compositional nature of the scene is available, we can design the ProbLog program to encode only the label $y$, and VAEL collapses to CCVAE. Whenever we can access extra information to map $y$ to a more fine-grained symbolic representation, we can inject this knowledge into the model through a ProbLog program. VAEL exploits the additional information to improve task generalization and data efficiency, as we will show in this section. Conversely, CCVAE is not able to exploit all the available information, thus requiring more training data to perform comparably and preventing reaching the same level of task generalization."*
> > >
> > > We hope the statement above clarifies the discussion on extra information and the comparison between CCVAE and VAEL.
> > > We would be happy to answer any additional questions.

---

> > > > ### Comment · Reviewer_2dcW · 2022-08-08
> > > > **Reply to Authors**
> > > >
> > > > I would like to thank the authors for their efforts in clarifying the extra information used by their method. In this case, I would be glad to keep my original score as "6: Weak Accept".

---

### Official Review · Reviewer_aUPz · 2022-07-09

**Rating:** 4
**Confidence:** 3
**Soundness:** 2 fair
**Presentation:** 2 fair
**Contribution:** 2 fair

**Summary:**

In this work, the author proposed to unify VAE with probabilistic logic programming. In doing so, the authors modify the generation part in VAE such that one part of latent space is used for a probabilistic logic programming which produced the decoded logic labels. To demonstrate the usefulness of the proposed method, the authors conduct experiments in two experiments: one of two MNIST images with sum as label and one of two Mario levels with the move as the label.

**Questions:**

Besides the weaknesses identified above, there are some other technical questions:
- how to sample given evidence (the details of the sampling process that corresponds to Eq 3)?
- How CCVAE is applied to the tasks given in the experiments? This is not straightforward as the original CCVAE paper uses different tasks.
- Regarding sampling: Can we sample conditioned on not only y but part of x? For example, can we sample also given the first digit (task 1) or the initial level before move (task 2)?  This may be more interesting if the tasks are made harder (see discussion of weaknesses above for the hardness of the tasks.)

**Limitations:**

No issue found.

**Strengths And Weaknesses:**

# Strengths
This work propose to unify unify VAE with probabilistic logic programming, which is an original problem to investigate. Although the proposed method is still an early-stage one (the experiments are limited and the proposed method should have been better motivated), it shows interesting property where the label from probabilistic logic programing can be more explicitly modeled, which shows some significance.

# Weaknesses

However this paper in its current form still have several weaknesses, which mostly happening for the experiments:

- It's unclear why experiments are designed in this way. Maybe some better motivation explaining the challenging / the goal of the proposed tasks are helpful.
- It's unclear how hard the proposed tasks are. On the one hand, comparing with (a wide range of) conditional generative model in the proposed tasks are missing. On the other hand there is no failure mode for the proposed method, which do not provides the readers with an understanding of the limitation of the proposed method.
- Regarding the "hardness" of the tasks: To which degree a extending of the current tasks makes the proposed method fail? For example, how many digits in a single image (task 1) / how many moves between two snapshot of levels (tasks 2).


These weaknesses make it hard for a reader to draw inspiration from this work.

# Summary after the rebuttal

I thank the authors for the substantial rebuttal. Still I would like to keep my score of 4 (see discussion below for details). I would encourage the author to concisely articulate the clarification/rebuttal in a revised version of the paper to strengthen the paper.

---

> ### Author Response · Authors · 2022-08-02
> **Answer to Reviewer aUPz**
>
> We thank the reviewer for their constructive comments. In the following sections, we describe the reasons behind the design of the proposed tasks and the challenges they represent. We also discuss more in detail the main limitation of our approach, mentioned in Section 6 of our paper. Finally, we reply to the technical questions.
>
> ## Motivations
> Both tasks present a compositional scene that allows us to emphasize the advantages of using a neuro-symbolic approach to logically reasoning upon the scene components. Moreover, the compositional nature of our tasks allows us to design completely novel tasks that imply arbitrarily complex forms of reasoning among the scene components. For example, in 2-digit MNIST, we can define not only the addition but also any other operation between two digits. Similarly, 2-state MARIO can be extended to trajectories of arbitrary lengths. This aspect makes the proposed tasks suitable for testing the generalization capability of the models.
> ## Hardness
> Despite their simplicity, the two tasks are yet challenging for a pure neural conditional generative model, like the state-of-the-art CCVAE, which cannot rely on the expressivity and strength of Probabilistic Logic Programs (PLPs). Indeed, in both tasks, the label can be mapped to a more fine-grained symbolic representation (e.g., from the addition to the pair of digits in 2-digit MNIST or from the move to the initial and final states in 2-state MARIO). However, this mapping is not deterministic since we may have multiple pairs of digits that sum to the same value or multiple pairs of states for a given move. Without a PLP, CCVAE has to learn the mapping in a subsymbolic way, resulting in scarce performances.
>
> ## Limitations
> VAEL inherits all the advantages of PLPs in terms of logical reasoning under uncertainty and differentiability. However, it also inherits one main limitation, i.e., their scalability. PLP inference is a \#P-hard problem with no known bounds. Scaling PLP inference is an open research question whose investigation is out of the scope of this paper. In this work, we are interested in showing that probabilistic logic programming can be tightly integrated with deep generative models, bringing new unprecedented capabilities to both worlds.
>
> ## Technical Questions
> *"How to sample given evidence (the details of the sampling process that corresponds to Eq 3)?"*
>
> By using conditional inference, ProbLog is able to compute the probability of the worlds given an evidence, namely $P(w_F | E;p)$ (e.g., the probability of a certain pair of digits given the sum). Then, we sample a single world from $P(w_F | E;p)$ by using Gumbel-Softmax [1]. The sampled world is consistent with the evidence $E$, since the distribution $P(w_F | E;p)$ is zero for those worlds that do not satisfy $E$.
>
> *"How CCVAE is applied to the tasks given in the experiments? This is not straightforward as the original CCVAE paper uses different tasks."*
>
> In the two tasks (2-digit MNIST and 2-state MARIO), CCVAE and VAEL are trained on the same dataset without any additional label. Both the models aim at maximizing the joint distribution $P(X,Y)$, where $X$ is the input image and $Y$ is its label.
> With respect to the original paper, we slightly modify the encoder and decoder mapping functions of CCVAE (see Appendix D.2 for more details) to ensure direct supervision on the latent space components in our experimental setting. The original CCVAE graphical model is preserved. Thus, in both tasks, the only difference between CCVAE and VAEL is that the latter presents a probabilistic logic program, which is exactly the method we propose to the community to inject domain knowledge into the generative model (e.g., the concept of addition between two digits in 2-digit MNIST task and the concept of moving along the four cardinal directions in 2-state MARIO task).
>
> *"Regarding sampling: Can we sample conditioned on not only $y$ but part of $x$? For example, can we sample also given the first digit (task 1) or the initial level before move (task 2)? This may be more interesting if the tasks are made harder (see discussion of weaknesses above for the hardness of the tasks.)"*
>
> We thank the reviewer for their question, which allows us to point out one of the advantages of using a probabilistic logic programming language. In ProbLog, we can easily switch from one kind of inference to another at run-time without retraining the model. For example, in 2-digit MNIST task, during the training, we sample conditioned on the label only. Then, at inference time, we can condition the sampling also on the value of one of the two digits by simply providing the value as evidence to the program. The same applies to 2-state MARIO task, where we can condition not only on the move (i.e., the label $Y$) but also on the starting or ending level.
>
> [1] E. Jang, et al. “Categorical Reparameterization with Gumbel-Softmax”. In: ICLR 2017.

---

> > ### Comment · Reviewer_aUPz · 2022-08-07
> > **Response to the rebuttal**
> >
> > I would like to thank the authors for the rebuttal that is very substantial (great job!), and with a good amount of effort in making a clear motivation.
> >
> > However my major concerns are not well addressed:
> > - The motivation is still seemingly weak without experiments/examples showing the case.
> > - My concerns on the hardness and the limitation of the proposed task/method are meant to be revealed by possible alternative models, rather than the theoretical complexity mentioned in the rebuttal  that the simple tasks in the paper are far from touching.
> >
> > To conclude, I would like to keep my score of 4. I would encourage the author to concisely articulate the clarification/rebuttal in a revised version of the paper, in which case the work would be strengthened.

---

> > > ### Author Response · Authors · 2022-08-08
> > > **Reply to Reviewer aUPz**
> > >
> > > We thank the reviewer for acknowledging and clarifying the nature of their concerns.
> > >
> > > We understand and respect the position of the reviewer. However, we believe that our experiments highlight the capabilities of our neuro-symbolic integration and showcase the motivations behind the design of the proposed tasks. In both tasks, we can find three core aspects: (i) compositionality, (ii) reasoning, and (iii) generalization. For example, the compositionality of 2-digit MNIST is represented by the two single digits themselves, while the reasoning is the stochastic symbolic mapping from the two digits to their sum; the generalization is performed at test time by logically changing the mapping from the addition to other mathematical operations, as shown in our experiments. Similarly, we can find the same three core aspects in MARIO task.
> > > We will include an additional statement to clarify the design of the tasks on line 173:
> > >
> > > *Both datasets present a compositional scene that showcases the advantages of using a neuro-symbolic approach to logically reasoning upon the scene components. Moreover, the compositional nature allows us to design novel tasks that imply arbitrarily complex forms of reasoning among the scene elements, thus testing the generalization capability of the models.*
> > >
> > > Regarding the second reviewer's concern, to the best of our knowledge, there are no alternative models able to exploit the same symbolic information to achieve a level of task generalization and data efficiency similar to VAEL. We would appreciate it if the reviewer could kindly provide us with some examples of alternative models they referred to in their comment.

---

### Official Review · Reviewer_t873 · 2022-07-11

**Rating:** 6
**Confidence:** 3
**Soundness:** 3 good
**Presentation:** 3 good
**Contribution:** 2 fair

**Summary:**

The presented paper bridges the gap between the idea of a VAEs and its applicability to neuro-symbolic systems that aim at pushing towards next generation systems that combine low level processing and high level reasoning capabilities of Deep Learning and Symbolic methods respectively. To this end, the authors derive a VAE model that internally makes use of a Probabilistic Logic Program, alongside all important components such as inference model and training objective. The authors provide an extensive empirical accounts of the proposed model.

**Questions:**

The questions are derived mostly from aspects mentioned in the "Weaknesses" section. I'd invite the authors to answer these questions to overcome the observed weaknesses for improving the paper to ultimately increase the chances of both contribution and visibility to the community. The list of questions is unordered and they range in quality and speed of expected answer (some are minor, quickly answerable whereas others are more fundamental, crucial, maybe difficult to answer):

* How can we ease modelling assumptions for the ProbLog component?
* What would be arguments against a GAN-style formalization in a similar fashion to VAEL?
* How far can the generalization capabilities of VAEL reach, what is its "ceiling"?

**Limitations:**

No contradictions or any sort of relevant mistake have been detected in the paper. The general clarity of the paper is an advantage. Existing bodies of work are being referenced excellently.

Sufficient details for reproduction are being provided alongside actual code which was not validated however. Societal impact is not being discussed IMHO not necessary to begin with for this concrete research project.

**Strengths And Weaknesses:**

The presented paper bridges the gap between the idea of a VAEs and its applicability to neuro-symbolic systems that aim at pushing towards next generation systems that combine low level processing and high level reasoning capabilities of Deep Learning and Symbolic methods respectively. To this end, the authors derive a VAE model that internally makes use of a Probabilistic Logic Program, alongside all important components such as inference model and training objective. The authors provide an extensive empirical accounts of the proposed model.

Acknowledging the difficulty of both investigating but also presenting it, still, the lack of theoretical properties to be characterized for the VAEL is certainly a short coming. Furthermore, aspects like computational complexity the authors might want to discuss for methods which should carry practical implications like VAEL. Also, IMHO it would greatly benefit the work to focus more on discussing the key assumptions/aspects involving the ProbLog component within VAEL. I'd also invite the authors to take the opportunity of extending their empirical analysis to a more ablative study of key components of their proposed model, such that the schematic differences presented in Fig.1 could be made more clear in the context of actual experimentation.

---

> ### Author Response · Authors · 2022-08-02
> **Answer to Reviewer t873**
>
>
> We thank the reviewer for their positive assessment and for raising interesting questions that allow us to better explain the strengths and limitations of our work.
>
> ### Technical Questions
> *"How can we ease modelling assumptions for the ProbLog component?"*
>
> We would like to point out that we believe that the human intervention in modeling the problem via ProbLog, is an important advantage of our method rather than a limiting factor we need to ease. In our work, we show that the capability of VAEL to use such information provides better control over the generation process than the state-of-the-art CCVAE. We also show that such control becomes fundamental in generalizing to new tasks. Moreover, by using ProbLog as the frontend, the user is extremely facilitated. Indeed, ProbLog is based on definite clause logic, a subclass of first-order logic. Therefore, conversely to propositional logic, we can express very compact rules without specifying the grounding over the elements of the Herbrand base. This aspect allows the user to define some "high-level" formal specifications that the neural components must satisfy.
>
> *"What would be arguments against a GAN-style formalization in a similar fashion to VAEL?"*
>
> We thank the reviewer for suggesting this comparison: the integration between probabilistic logic programming (PLP) and other families of generative models is a very interesting future research direction. However, this work focused on integrating PLP and variational autoencoders.
>
> *"How far can the generalization capabilities of VAEL reach, what is its "ceiling"?"*
>
> We thank the reviewer for their question, which allows us to better clarify the strengths and limitations of our approach. Considering the 2-digit MNIST dataset, VAEL is trained on the addition and is able to generalize to any other mathematical operation between two digits. Regarding MARIO dataset, once trained on two-state trajectories, VAEL can generalize to trajectories of arbitrary length. To the best of our knowledge, this level of generalization cannot be achieved by any other generative model. However, VAEL generalization capability is limited in terms of scalability. This is due to the probabilistic logic component, which does not allow VAEL to scale up to more computationally demanding tasks (e.g., a large number of digits or a wider MARIO grid world). We would like to point out that solving the known scalability issues of ProbLog is not a key point in this work. However, it represents a promising future research direction to further extend the generalization capability of VAEL.

---

> > ### Comment · Reviewer_t873 · 2022-08-06
> > **Thank You**
> >
> > The authors answered the questions raised by the review. The initial positive assessment remains. Out of curiosity, regarding the author's opinion regarding Q2, I'd like to rephrase the question in the following way: What are the aspects of the design choices you had to take for your VAEL setup that are specific to the VAE generative model class?

---

> > > ### Author Response · Authors · 2022-08-06
> > > **Reply to Reviewer t873**
> > >
> > > We thank the reviewer for confirming their positive assessment. Regarding their interesting question, we can distinguish between model and network architecture choices.
> > > Regarding the model, we had to devise the correct probabilistic graphical model that balances simplicity while ensuring the conditional independence assumptions.
> > > In terms of the architecture, we retained the architecture of VAE as much as possible. Along with the encoder and decoder networks, we introduced a MLP network to interface the latent vector of the model with ProbLog. All details on the architectures are reported in Appendix D.
> > >
> > > Please, let us know if there are any additional questions; we would be happy to answer them.

---

### Official Review · Reviewer_icZs · 2022-07-12

**Rating:** 5
**Confidence:** 4
**Soundness:** 3 good
**Presentation:** 3 good
**Contribution:** 2 fair

**Summary:**

This paper proposes a method VAEL, which combines VAE and ProbLog. The main idea is to interpret a subset z_sym of the latent representation z of VAE as symbolic representation of the input. A neural network maps z_sym to probabilities of facts in a ProbLog program. VAEL is tested on two image generation tasks and is shown to have better generalization.

**Questions:**

NA

**Limitations:**

Yes

**Strengths And Weaknesses:**

Strengths:

-- The proposed method is intuitive and is a reasonable approach to link VAE and symbolic reasoning.

-- Experimental results shows advantage over competitor CCVAE: VAEL has better generalization and requires less training samples. Conditional image generation and task generalization (replacing sum with multiplication etc) are also demonstrated, which cannot be easily done with CCVAE.


Weaknesses:

-- The proposed method requires human intervention to explicitly write the ProbLog program. For example, in the 2-digit MNIST task, a human specifies that the label is the sum of the two digits. This requirement limits the applicability of the proposal.

-- The two tasks (2-digit MNIST and 2-state Mario), although interesting, do not involve uncertainty: if the two sub-images are labeled, the overall label is uniquely determined. In other words, the uncertainty in the ProbLog program only exists in the correctness of classification of the two sub-images. This represents a limited case of reasoning with uncertainty.

---

> ### Author Response · Authors · 2022-08-02
> **Answer to Reviewer icZs**
>
> We thank the reviewer for the time spent reading our paper and for the thoughtful comments.
>
> ### Weaknesses
> *"The proposed method requires human intervention to write the ProbLog program explicitly. For example, in the 2-digit MNIST task, a human specifies that the label is the sum of the two digits. This requirement limits the applicability of the proposal."*
>
> We thank the reviewer for pointing out this requirement. Let us clarify this aspect as we believe human intervention strengthens our method.
> To the best of our knowledge, we are the first to propose a viable method to inject knowledge into a generative autoencoder model.
> From a generative viewpoint, this knowledge is nothing more than a formal specification of the desired property of the scene.
> By encoding this knowledge in the ProbLog program, we provide additional control of the generation process, as well as the capability of generalizing to novel tasks without retraining. Therefore, unlike classical deep generative models, which cannot fully exploit the domain knowledge available at training time, VAEL is able to i) perform better than the state-of-the-art CCVAE, ii) generalize to novel tasks, and iii) achieve better performance in contexts characterized by data scarcity.
> Moreover, we would like to point out that there are no limitations to the applicability of VAEL. In all those scenarios where no information is available at training time, we can design the ProbLog program to encode only the label $Y$, and VAEL collapses to a CCVAE. Thus, the proposed method can be applied to any classical conditional generative task. Furthermore, ProbLog uses first-order-logic syntax, which is at the same time very expressive but extremely compact, thus reducing the burden of human intervention.
>
> *"The two tasks (2-digit MNIST and 2-state Mario), although interesting, do not involve uncertainty: if the two sub-images are labeled, the overall label is uniquely determined. In other words, the uncertainty in the ProbLog program only exists in the correctness of classification of the two sub-images. This represents a limited case of reasoning with uncertainty."*
>
> The point raised by the reviewer could be analyzed in terms of the difference between epistemic uncertainty, i.e., the uncertainty in the model (e.g., the uncertainty over the parameters of a classifier) and aleatoric uncertainty, i.e., uncertainty in the task/data (e.g., the uncertainty over the roll of a dice). We do not consider aleatoric uncertainty in the designed tasks. However, our generative tasks strongly rely on (digit / levels) classification capabilities of our model. Thus, reasoning over the class probabilities is fundamental for enabling learning in our system. Modeling the same epistemic uncertainty has been critical in learning  (discriminative) neural probabilistic logic programs like DeepProbLog. VAEL brings such capabilities to the generative settings, showing the advantages of the proposed neuro-symbolic integration in two tasks that, despite their simplicity, are yet challenging for the state-of-the-art CCVAE. We thank the reviewer for the interesting comment; we will add this consideration to the paper.

---

> > ### Comment · Reviewer_icZs · 2022-08-06
> > **Thank you for the response**
> >
> > The two issues still stand, but I agree with some of the authors' points on them. I've increased the rating to 5.

---

### Meta-Review · Area_Chair_JnDi · 2022-08-25

**Recommendation:** Accept
**Confidence:** Less certain

**Metareview:**

The paper proposes to include a component enforcing logical constraints on top of a variational autoencoder (VAE). The resulting method, VAEL. does so by leveraging ProbLog in addition to a neural encoder and decoder. VAEL is employed for simple generative tasks with constraints over the outputs, such as conditional image generation with MNIST and small Mario levels.

The reviewers have appreciated the direction where VAEL is heading and the importance of integrating constraints in deep generative models. Some concerns are still open. Specifically, the motivation behind some architectural choices (and the specific choice of VAEs as deep generative models) and the small-scale nature of the experiments. The complexity and scaling campabilities of performing symbolic reasoning with ProbLog are not discussed in depth.


**Award:**

No

---

### Decision · Program_Chairs · 2022-09-14

Accept